# Preparation and Characterization of Cellulose Nanocrystals from Bamboos and Their Application in Cassava Starch-Based Film

**DOI:** 10.3390/polym15122622

**Published:** 2023-06-08

**Authors:** Parichat Thipchai, Winita Punyodom, Kittisak Jantanasakulwong, Sarinthip Thanakkasaranee, Sasina Hinmo, Kanticha Pratinthong, Gopinath Kasi, Pornchai Rachtanapun

**Affiliations:** 1Doctor of Philosophy Program in Nanoscience and Nanotechnology (International Program/Interdisciplinary), Faculty of Science, Chiang Mai University, Chiang Mai 50200, Thailand; parichat_thi@cmu.ac.th; 2Department of Chemistry, Faculty of Science, Chiang Mai University, Chiang Mai 50200, Thailand; winitacmu@gmail.com; 3Center of Excellence in Materials Science and Technology, Chiang Mai University, Chiang Mai 50200, Thailand; kittisak.jan@cmu.ac.th (K.J.); sarinthip.t@cmu.ac.th (S.T.); 4Division of Packaging Technology, School of Agro-Industry, Faculty of Agro-Industry, Chiang Mai University, Chiang Mai 50100, Thailand; gopiscientist@gmail.com; 5Cluster of Agro Bio-Circular-Green Industry (Agro BCG), Chiang Mai University, Chiang Mai 50100, Thailand; 6Master of Science Program in Physical Chemistry, Faculty of Science, Chiang Mai University, Chiang Mai 50200, Thailand; hinmo7231@gmail.com (S.H.); kanticha_p@cmu.ac.th (K.P.)

**Keywords:** bamboo, biopolymer, cassava starch, cellulose nanocrystals, composite film, ultrasonication

## Abstract

Cellulose from different species of bamboo (*Thyrsostachys siamesi* Gamble, *Dendrocalamus sericeus* Munro (DSM), *Bambusa logispatha*, and *Bambusa* sp.) was converted to cellulose nanocrystals (CNCs) by a chemical–mechanical method. First, bamboo fibers were pre-treated (removal of lignin and hemicellulose) to obtain cellulose. Next, the cellulose was hydrolyzed with sulfuric acid using ultrasonication to obtain CNCs. The diameters of CNCs are in the range of 11–375 nm. The CNCs from DSM showed the highest yield and crystallinity, which was chosen in the film fabrication. The plasticized cassava starch-based films with various amounts (0–0.6 g) of CNCs (from DSM) were prepared and characterized. As the number of CNCs in cassava starch-based films increased, water solubility and the water vapor permeability of CNCs decreased. In addition, the atomic force microscope of the nanocomposite films showed that CNC particles were dispersed uniformly on the surface of cassava starch-based film at 0.2 and 0.4 g content. However, the number of CNCs at 0.6 g resulted in more CNC agglomeration in cassava starch-based films. The 0.4 g CNC in cassava starch-based film was found to have the highest tensile strength (4.2 MPa). Cassava starch-incorporated CNCs from bamboo film can be applied as a biodegradable packaging material.

## 1. Introduction

Nowadays, people are becoming more environmentally conscious due to the effects of global warming and they demand new environmentally friendly materials that might take the place of materials derived from fossil fuels [1]. The most abundant organic substance in the world is cellulose, which is the main component of plant cell walls. Cellulose is composed of linear chains of glucose units connected by *β*-1-4-glycosidic bonds between the C-4 and the anomeric C-1 of the two sugar units [2]. Cellulose is derived from plants such as wood, bamboo, sugarcane bagasse, pineapple leaf, etc. [3]. Bamboo is one of the most attractive sources and is commonly cultivated in tropical and subtropical areas (Southeast Asia), where it grows naturally [4]. In comparison to most wood, bamboo grows more quickly. Bamboo has a considerably shorter harvesting period, a high yield, and a low cost [5]. In addition, bamboo also grows in other regions such as Africa, South America, and the Caribbean, and each region’s bamboo has unique properties based on species, climate, and soil conditions [6]. Bamboo culm is composed of vascular bundles embedded in parenchyma tissue. Because of bamboo’s strength, durability, and flexibility, it is frequently used for construction, furniture, and handicrafts. The strength of bamboos also depends on age ranges and species [7,8,9]. Each species of bamboo in Southeast Asia is used for different purposes. For example, the bamboo shoots species *Bambusa* sp. is edible, whereas *Thyrsostachys siamesi* Gamble and *Dendrocalamus sericeus* Munro are used for furniture and *Dendrocalamus sericeus Munro* is used for basketry [10]. In general, there are many species of bamboo, and each species has a different size, shape, and composition of bamboo [11,12]. Cellulose from bamboos is frequently used as a reinforcement for polymer composites due to its high fiber content and exceptional tensile strength. As a result, the fibers obtained from bamboos are thought of as potential raw materials for the production of cellulose [13].

In recent years, numerous studies have investigated various chemical processes that extract cellulose materials into sizes ranging from millimeters to nanometers. Acid hydrolysis is commonly used to remove the amorphous parts of fiber structures, leaving behind the crystalline parts of cellulose, producing cellulose nanocrystals (CNCs) [14]. CNCs can be extracted from various natural fibers such as sugarcane bagasse [15], walnut shell [16], palm oil empty fruit bunch [17], garlic stalks [18], and bamboo [19,20,21]. The extraction of CNCs from different bamboo species using chemical methods has been reported. Bosenbecker et al. successfully extracted CNCs from bamboo (*Bambusa vulgaris*) by acid hydrolysis [22]. The same method was used to extract the nanocellulose from *Oxytenanthera abyssinica* (Ethiopian lowland bamboo). The result showed a higher crystallinity index and good thermal stability [23]. Do et al. extracted CNCs from bamboo (*Bambusa blumeana*) fibers with an average diameter of 2 to 10 nm and a high aspect ratio by the chemical–mechanical method [24]. Typically, CNCs have an average size of 100–300 nm in length and 5–20 nm in diameter, with a rod-like shape due to the ordered arrangement of cellulose chains. CNCs have rod-shaped nanoparticles derived from the acid hydrolysis of cellulose fibers. CNCs possess unique physicochemical properties of high aspect ratio, large surface area, and high crystallinity [25,26]. CNCs have the advantage of excellent crystallinity, resulting in great strength and strong thermal stability. In addition, CNCs are a biodegradable, lightweight, low-density, and transparent material [27,28]. Moreover, CNCs possess an abundance of reactive hydroxyl groups on their surface, making them an ideal reinforcement filler for biopolymer composites. These properties improve reinforcement for biopolymer composites [29]. Because of their outstanding properties, CNCs have been used in various applications (e.g., packaging, foods, cosmetics, pharmaceutical, and biomedicine) [30]. In the development of packaging films, natural and biodegradable polymers have been widely researched. Among biopolymers, starch is one of the most interesting options in film development because of its biodegradability, renewability, and low price. Cassava (*Manihot esculenta*) is an important commercially cultivated crop in Thailand as it is a low-priced starch material [31]. The advantages of cassava starch are its availability, affordability, and high viscosity, which results in it readily being cast into films [32].

Nevertheless, cassava starch film has poor mechanical characteristics due to its brittleness and poor moisture barrier [33]. Several research groups have improved such drawbacks by the introduction of other components such as clays, chitosan, and carboxymethylcellulose). CNCs are also an interesting component in the improvement of water vapor barriers and mechanical properties of cassava starch films. Several studies have demonstrated that adding CNCs efficiently enhances the characteristics of starch-based nanocomposite films [34].

Until today, several studies have reported the preparation of CNCs from bamboo fibers, but in most cases, the researchers have only used a few species of bamboo, namely, *Phyllostachys heterocycle, Dendrocalamus,* or *Pseudosasa amabilis* [20,21,35]. Thus, the present research aimed to study CNC properties extracted from different species of bamboo, namely, *Thyrsostachys siamesi* Gamble (TSG), *Dendrocalamus sericeus* Munro (DSM)*, Bambusa longispatha* (BL), and *Bambusa* sp. (BS). The chemical constituents of the raw materials were analyzed. The yield, size, morphology, and chemical structure of cellulose and the CNCs were derived from different bamboo species. Among them, the DSM sample exhibited the highest yield and better crystallinity in CNCs. To the best of our knowledge, this is the first report of a comparative assessment of four different bamboo species used for CNC synthesis. Different amounts of CNC-DSM were added to the cassava starch-based films, and their physicochemical and water-resistant properties were analyzed.

## 2. Materials and Methods

### 2.1. Materials

The culm of four different bamboo species (TSG, DSM, BL, and BS) was obtained from Samoeng District, in Chiang Mai Province, Thailand. The sodium hydroxide, acetic acid, sodium chlorite, and sulfuric acid were purchased from Merck & Co., Inc., Darmstadt, Germany. All chemicals were reagent grade and used without further purification.

### 2.2. Raw Materials Preparation

The outer layer of bamboo culm was peeled out. Then, the bamboo culm was cut into pieces (2 × 6 inches). The pieces of bamboo were dried in a hot air oven at 85 ± 5 °C for 12 h (Universal oven UN30, Memmert GmbH Co. KG, Schwabach, Germany). The dried pieces were ground into powder using Grinder (Grinder ML-SC5-III, Ming Lee Industrial Ltd., Hong Kong, China). Next, the powder was crushed with the high-speed blender (Dxfill machine, DXM-700-F, Shanghai, China) at 35,000 rpm for 15 min, then passed through the 250-micron sieve. In the next step, the bamboo powder was dried in a hot air oven at 85 ± 5 °C for 12 h and stored in a desiccator to maintain constant moisture. According to the Technical Association of the Pulp and Paper Industry standard or TAPPI techniques T204 om-88, T203 om-88, T222 om-88, and T211 om-93, respectively, the extractive, holocellulose, hemicellulose, α-cellulose, lignin, and ash contents of the raw materials were assessed. Briefly, the extractives in bamboo fiber were determined; 10 g of the sample powder, ethanol, and benzene were combined and heated to 80 °C for 5 h. Then, the sample (1 g) was hydrolyzed using 72% sulfuric acid to evaluate its lignin content. The lignin was then filtered, cleaned, and dried in a hot air oven at 85 ± 5 °C for 12 h. The extractive-removed fiber was used for holocellulose analysis. The holocellulose including α-cellulose and hemicellulose was analyzed from extractive-removed fiber using the acid chlorite method according to Browning [36]. The hemicellulose was removed from the holocellulose using 17.5% *w*/*v* sodium hydroxide at 25 °C to obtain α-cellulose. The ash content was estimated using sample powder (1 g) that had calcined at 575 °C for 3 h, as shown in Figure 1. Every measurement of the sample was analyzed using triplicate.

### 2.3. Preparation of Cellulose from Bamboo Fiber

The first method was chemical defibrillation to remove hemicellulose, lignin, and other non-cellulosic substances from microfibrils (Figure 2). The raw bamboo powder (50 g) was boiled into 1000 mL of sodium hydroxide (18% *w*/*v*) solution at 80 °C under continuous stirring at 500 rpm speed for 5 h (Overhead Stirrers, Bethai Bangkok Equipment & Chemical Co., Ltd., Bangkok, Thailand). Then, the product was filtered and washed with distilled water until the pH became 6.5–7. Next, the pulp was dried at 85 ± 5 °C, for 12 h. The sodium chlorite bleaching to obtain cellulose was prepared using conventional methods according to Pacaphol and Aht-Ong [37]. The dried pulp (50 g) was bleached by 500 mL sodium chlorite (3.4% *w*/*v*) and acetate buffer (the mixture between 452 mL of sodium hydroxide (5.4% *w*/*v*) and 75 mL acetic acid (15% *w*/*v*). The bleaching process was conducted at 85 ± 5 °C for 12 h under continuous stirring at 500 rpm speed. Later, the product was filtered and washed with distilled water until the pH became 6.5–7.0. Then, cellulose was dried at 85 ± 5 °C for 12 h. The entire bleaching procedure was repeated 2 times, resulting in greater whiteness of the bamboos, then cellulose was kept in a desiccator.

### 2.4. Extraction of CNCs from Different Bamboo Fibers

The CNC was extracted according to Mandal and Chakrabarty [15]. The experimental design is shown in Figure 2. The cellulose 50 g was acid-hydrolyzed with 1500 mL sulfuric acid (32% *v*/*v*) for 5 h at 50 °C under continuous stirring at 500 rpm speed. The hydrolysis was quenched by adding excess distilled water (500 mL) until the pH was 2.3–2.6. Then, the suspension was neutralized with sodium hydroxide (5% *w*/*v*) until the pH became 6.5–7.0 [38]. Whereupon, the suspension was spun by using the high-speed blender (High-performance blender, Smart-ID Group Ltd., Anitech, Nonthaburi, Thailand) at 36,000 rpm speed for 15 min. The gel was soaked overnight, then the sample speed was spun at 36,000 rpm for 15 min again. The gel CNC was ultrasonicated at 20 kHz frequency with an output power of 500 W by an ultrasonic generator (Sonics & Materials, INC. 53 Church Hill RD. Newtown, CT, USA) in an ice bath at 25 ± 3 °C for 30 min. Next, the gel CNC was centrifuged (HERMLE Labortechinik GmbH, Wehingen, Germany) at 6000 rpm for 10 min, and the sample was repeated until the precipitate generation was terminated. The supernatant was sonicated for 30 min in an ice bath again to avoid overheating. The CNC suspension was stored in a refrigerator at 4 °C. The sample (500 mL) was frozen in a freeze-dryer (DW-10N Freeze Dryer, Chongqing Drawell Instrument Co., Ltd., Chongqing, China) at a temperature of −40 °C with a vacuum at 20 Pa for 12 h [39]. The percent yield of CNC from bamboos was determined by calculating from Equation (1):(1)$$\mathrm{Yield}\;\left(\%\right)=\frac{\;\left(\mathrm{mass}\;\mathrm{of}\;\mathrm{weight}\;\mathrm{of}\;\mathrm{freeze}-\mathrm{dried}\;\mathrm{CNC}\right)}{\left(\mathrm{mass}\;\mathrm{of}\;\mathrm{cellulose}\right)}\times 100$$

### 2.5. Preparation of Nanocomposite Films

The nanocomposite films reinforced with CNCs were prepared according to the method of Mandal and Chakrabarty with minor modifications [15]. In Figure 2, the experimental design is displayed. In brief, the cassava starch (5.0 g) was boiled in 50 mL of distilled water at 80 °C under constant stirring for 15 min, while CNC 50 mL (0.2, 0.4, and 0.6 g) were ultrasonicated at 25 °C for 15 min. Both were mixed under constant stirring for 10 min. Then, glycerol (1.5 g) was added to the mixture. The obtained nanocomposite solution was cast on the plate and dried in the hot air oven at 45 °C for 24 h.

### 2.6. Characterizations

The functional groups of sample powders (raw bamboos, pulps, cellulose, and CNCs) were determined by a Fourier Transform Infrared Spectrometer (FTIR Spectrometer, FT/IR-4700, JASCO International Co., Ltd., Pfungstadt, Germany). The sample (~2 mg) with KBr was used to make pellets for measurement at a wavenumber range of 4000–500 cm^−1^, with a resolution of 4 cm^−1^. In addition, the FTIR spectrometer in ATR mode was used to examine the structural connections of nanocomposite films reinforced with CNCs. The film samples were cut into small pieces (10 mm × 10 mm), then placed in the sample container. The spectra were run using a scan rate of 64, in the wavenumber of 4000–500 cm^−1^.

The morphology of raw bamboos and cellulose was analyzed using a scanning electron microscope (SEM) (Phillip XL 30 ESEM, FEI Company, Hillsboro, OR, USA) with a large field detector. The acceleration voltage was 15 kV at the 100× and 500× magnifications.

A field emission scanning electron microscope (FE-SEM) with a STEM function was used to characterize the morphology of CNCs (FE-SEM, JSM-IT800, JEOL, Peabody, MA, USA) at 20 kV voltage with 30,000× magnification. A concentration of 0.01% *w*/*v* of the CNCs was dispersed in distilled water in the beaker and sonicated for 30 min. The sample (10 µL) was dropped on a Cu grid and dried using light for 15 min. The image-J software was used to measure the length and width of raw bamboos, cellulose, and CNCs.

The particle size of CNCs was measured by laser diffractometry using a Nano Size Particle Analyzer (Zetasizer, Malvern Panalytical Ltd., Malvern, Worcestershire, UK) in the range between 0.6 and 6000 nm. The average particle size was calculated using the software (DTS, version 5.00 from Malvern Panalytical Ltd., Malvern, Worcestershire, UK) after thirteen measurement cycles lasting 10 s each [15].

The cellulose and CNCs from four bamboo samples were analyzed using an X-ray diffractometer (JEOL JDX-80-30 X-ray diffractometer, Brucker, Regina, SK, Canada). The scattering angle (2θ) ranged from 5 to 50° at a scan rate of 2° min^−1^. The crystallinity index (CrI) was determined by using an empirical Equation (2). The I_Cr_ is the crystalline phase’s integrated intensity, whereas I_non_–_Cr_ is the non-crystalline phase’s integrated intensity as a background as follows [40]:(2)$$\mathrm{Crystallinity}\;\mathrm{Index}\;\left(\%\right)=[\left(I_{\mathrm{Cr}}/\left(I_{\mathrm{Cr}}+{\;I}_{\mathrm{non}-\mathrm{Cr}}\right)\right]\times 100\;$$

The crystallite size was calculated according to Scherer’s Equation (3):(3)$$D\;=\frac{k\lambda }{\beta cos\theta }\;$$
where *k* is a constant with a value *(k* = 0.94), *λ* = 0.15406 nm, and *β* is the full width at half maximum of 200 reflections [41].

### 2.7. Characterization of Nanocomposite Films

The tensile tester (MCT-1150 model, A&D company, Tokyo, Japan) was used to determine the tensile strength (TS) and elongation at break (EB) of the cassava starch-CNC-DSM nanocomposite films at a crosshead speed of 10 mm/min. The films were cut into 5 mm × 50 mm. All samples were equilibrated in the desiccator at 50 ± 10% RH 25 °C for 48 h and were analyzed according to the JISK-6251-7.

The water solubility of nanocomposite films was measured using a method modified by Rachtanapun et al. [2]. Initially, sections of the film samples (20 mm × 20 mm) were cut out and dried at 45 °C for 24 h, then stored in desiccators at 0% RH 25 °C for 48 h. The dried sample was weighed at 0.05 g initial dry weight. The dried sample was submerged in 50 mL of distilled water with shaking for 24 h. Thereafter, they were filled into No. 4 filter sheets and dried at 105 °C for 24 h. All dimensions were run in triplicate. The percentage of water solubility was calculated using the following Equation (4):(4)$$\mathrm{Solubility}\;\left(\%\right)=\frac{W_{i}-W_{f}}{W_{i}}\times 100$$
where W_i_ and W_f_ represent the weight of the initial sample before submersion and the weight of the final product after water soluble, respectively.

The percentage swelling of nanocomposite films was determined by adapting the method of Suriyatem et al. [42]. Briefly, all film samples (20 mm × 20 mm) were equilibrated in the desiccator at 0% RH 25 °C for 48 h. The samples were submerged in distilled water at 25 °C for 1 h. The swelled films were weighed after gaining weight, and Equation (5) was used to determine the swelling percentage. Every measurement was tested in triplicate.
(5)$$\mathrm{Swelling}\;\left(\%\right)=\frac{W_{1}-W_{0}}{W_{1}}\times 100$$
where W_0_ and W_1_ are pre- and post-water submerged weights of the films, respectively.

The water vapor permeability (WVP) of nanocomposite films was measured according to the ASTM standard method (ASTM, E96-93, 1993). In a nutshell, the films were cut into pieces and fit into cups that contained 10 g of dried silica gel and sealed with paraffin wax. Then, the sample was stored in desiccators before starting the test. Thereafter, the cup was placed at 55% RH, 25 °C. Daily weight measurements of the sample cup were recorded, and the WVP was calculated [43]. All sample tests were run in triplicate.

The surface of nanocomposite films reinforced with CNCs from bamboos was analyzed using an atomic force microscope (AFM) (NanoScope IIIa, Di digital instruments Veeco Metrology Group, Cambridge, UK). The nanocomposite films are cut into small pieces (10 mm × 10 mm), and the test is carried out on its surface. The average was performed using the program (Nanoscope III 5.12r3).

To analyze the hydrophilic property of nanocomposite films, the dynamic contact angle of nanocomposite films was assessed utilizing a drop of water (volume 10 µL) on the process samples (20 mm × 20 mm) by using a DSA30B Drop Shape Analyzer (KRÜSS, Hamburg, Germany). The contact angle on both sides of the drop formed was calculated at 0, 20, 40, and 60 s according to the method of Thanakkasaranee et al. [43]. The average of five measurements was used to compute the dynamic water contact angle of each film.

### 2.8. Statistical Analysis

All data were shown as averages with standard deviations. The significance of differences at the significance level of *p*-value < 0.05 was DSA30B Drop Shape Analyzer (KRÜSS, Hamburg, Germany) assessed using one-way ANOVA. The SPSS program version 16.0 was used to conduct statistical analysis (SPSS Inc., Chicago, IL, USA).

## 3. Results and Discussion

### 3.1. Chemical Constituents

The chemical constituents of raw bamboo fiber (TSG, DSM, BL, and BS) are shown in Table 1, and the calculations of such chemical constituents are presented in Appendix A. The raw bamboo fiber from four species consists of holocellulose from 64.4 to 74.6%, while α-cellulose content was approximately 37.4–42.5%. In addition, the fiber contains hemicellulose content of 27.1–32.1%, and there are other components, 22.5–28.9% lignin, 3.1–4.9% extractive, and 1.9–2.4% ash. The chemical constituent from the four species of bamboo is that BL has the most lignin, while TSG has the least. Raw bamboos have the least α-cellulose in BS and the most α-cellulose in DSM and hemicellulose content. The proportion of α-cellulose is strong with high crystallinity, but lignin and hemicellulose are amorphous regions. These can be removed by alkaline treatment and bleaching process [37]. The amount of α-cellulose affects the production of CNCs. The chemical constituents of the bamboos were observed to be slightly different because they are of a different species, which is consistent with previous research on different bamboo species [12,44,45]. This can be explained by analyzing the morphology of bamboo fiber cells and parenchyma cells, showing the morphology of several macerated bamboo fibers [39]. However, the amount of hemicellulose, lignin, and extractive in the initial fibers was effective for removal using the same process [46].

### 3.2. FTIR Analysis

The FTIR was used to analyze the chemical changes in the raw bamboos, pulps, cellulose, and CNC stages of the four different species of bamboo. Figure 3 shows the FTIR spectra of TSG, DSM, BL, and BS. Dotted lines highlight the different stages of the sample. The peak at 2902 cm^−1^ was ascribed to the stretching vibration of the cellulose C–H groups, while the peak at 3444 cm^−1^ was assigned to the O–H stretching vibration of the hydroxyl groups in the cellulose molecules [16]. However, there is a change from raw bamboos to cellulose, which brought about changes in the structure of samples, as observed from the FTIR spectra. The stretching vibration of C=O, which came from the ferulic and p–coumaric (lignin), was attributed to a shoulder peak at 1735 cm^−1^ in the spectra of raw bamboos. These peaks disappeared in the spectra of pulps and cellulose due to the removal of hemicellulose and lignin from chemical treatments. Furthermore, the absorbance peaks were at 1510 and 1425 cm^−1^ of the spectra of pulps and cellulose, which are C=C stretching from aromatic rings of lignin. The peak at 1639 cm^−1^ in the spectra of all the samples was attributed to the H–O–H stretching vibration of the adsorbed water due to the hydroxyl groups in cellulose [20]. The peak at 895 cm^−1^ is *β*-glucosidic linkages between the sugar units that are the C_1_–H deformation of cellulose [47]. The spectra at 1110 cm^−1^ were assigned to the C–O–C stretching of the anhydroglucose ring [48]. The peak around 1235 cm^−1^ was the C–O–C stretching of esters, while at 1160 cm^−1^, it was cuaiacyl C–H and syringyl C–H [49]. The peak at 1060 cm^−1^ represents the C–O stretching of hemicellulose or aryl-alkyl ether in lignin [17]. The spectra of (a) TSG, (b) DSM, (c) BL, and (d) BS showed no significant differences, indicating that the sulfuric acid hydrolysis process did not affect the characteristics of the cellulose molecular structure. However, the relative amount of cellulose in the sample increased due to the decrease in the amounts of other components during hydrolysis, which led to a slight increase in the intensity of the narrow peak at 1060 cm^−1^ from raw bamboos to CNCs. Moreover, the spectrum of CNC shows strong hydrogen-bonded (O–H stretching vibration) at 3600−3200 cm^−1^ of typical cellulose [50].

### 3.3. XRD and Percent Yield of Cellulose and CNC of Four Different Bamboo Species

The XRD patterns of typical cellulose and CNC are shown in Figure 4. In every instance, the CNC retained the original cellulose I crystal structure [51]. The XRD patterns demonstrate the distinct crystalline structures of cellulose and CNCs. The diffraction peaks for cellulose, which correspond to the (1–10), (110), (200), and (004) crystalline patterns, are found at 2θ = 15.4°, 16.4°, 22.5°, and 34.5°, respectively. These peaks are typical cellulose allomorphs I of parallel glucan chains [52]. The CNC peaks relate to the (1–10), (110), (200), and (004) crystalline patterns, which are typical cellulose allomorphs II structures, and they are apparent at 2θ = 12.5°, 20.2°, 22.3°, and 34.5° [53]. This demonstrates that the crystalline structure of CNCs was changed from type I to type II by the hydrolysis of cellulose by sulfuric acid, mechanochemical reactions, and ultrasonic processes. On the other hand, the crystallinity index or CrI (Table 2) of the CNC is calculated to be 38.9−42.2%, lower than that of cellulose (46.3–45.4%), where the CrI was calculated using a method of resolution of the peak. The results of this crystallinity index are in the same direction as the result of the percentage yield (Table 2) of cellulose and CNC, in descending order as follows: DSM (42.1% and 33.9%), TSG (40.8% and 32.9%), BL (41.5% and 31.4%), and BS (40.1% and 30.5%), respectively. The crystallite size of the cellulose from bamboos was 2.1–2.9 nm, whereas the crystallite size of the CNCs was 1.3–1.6 nm. The calculations are shown in Appendix A. However, the crystallite size of the cellulose and CNCs obtained in this study was relatively smaller than that of other works. The occurrence of smaller crystallite sizes of both the cellulose and CNCs obtained in this study might be due to twice bleaching and twice grinding after the bleaching. The crystallite size of cellulose from pineapple leaf fibers was 3.6–3.8 nm [54], whereas the crystallite size of cellulose from wood samples was 2.4–2.1 nm [55]. Moreover, the crystalline size of CNCs from *Malaysia indica* rice straw was 1.7 nm [56], whereas the CNCs from *Oxytenanthera abyssinica* (Ethiopian lowland bamboo) were 4.0 nm [23]. Rhim et al. reported that the crystallite size of cellulose from onion skin was 2.1 nm. In addition, Rhim et al. also investigated CNCs from onion skin extracted using different concentrations of 45, 55, and 65% sulfuric acid and found that the crystallite size of CNCs from onion skin was dependent on the concentration of sulfuric acid. The crystallite of size CNC from onion skin decreased with increasing the concentration of sulfuric acid, in which the crystallite size of CNC-45, CNC-55, and CNC-65 was 2.73, 2.43, and 1.76, respectively. This indicates that the higher concentration of sulfuric acid hydrolysis (strong acid) significantly affects the removal of amorphous domains, breaking the bundle of cellulose and forming a smaller crystallite of CNC [57]. Tang et al. reported that the breakdown of smaller crystallite size and an increase in CrI were caused by the alteration in crystallite size during acid hydrolysis [58]. The difference in CrI and crystallite size depends on the species of bamboo, which contain different components such as hemicellulose and α-cellulose, as shown in Table 1 [59]. From the experiment, it was found that the CrI of cellulose from all species was similar. The CrI of cellulose is the highest in DSM, and the BL is the lowest. Furthermore, the CrI of CNCs from DSM was the highest and BS was the lowest. Nevertheless, there is a noticeable decrease in the CrI of the CNCs made from several bamboo species as compared to the original cellulose. It was likely that the hydrolysis and probably the ultrasonication treatment disturbed the crystalline domains of the cellulose chains [60]. Moreover, the high shearing action during the high-speed blending process may result in damage either through breaking or peeling off of the cellulose chains on crystalline patterns [61]. However, less crystalline CNC II particles have better water dispersibility and greater material compatibility when compared to native cellulose. Furthermore, CNC II particles can achieve increased ductility in rigid polymeric matrices [62].

### 3.4. Visibility Changes in the Synthesized CNC

During the CNC synthesis process, visibility changes in the DSM sample are displayed in Figure 5. Figure 5a revealed that the raw material of DSM bamboos in the aqueous phase is brown-yellow. After the alkaline treatment, the sample of DSM exhibited a blackish-brown color due to the initial lignocellulosic material (Figure 5b). This is caused by the hydrolysis of starch, pectin, and hemicelluloses in the pulp fiber [63]. Figure 5c shows a white-color appearance due to the bleaching process. This color change is attributed to removing lignin and hemicellulose using sodium chlorite to oxidize lignin [64]. Subsequently, the sample was acid hydrolyzed and showed half-yellowish-white color with increased dispersion stability (Figure 5d). According to Lazko et al., the yellowing of cellulosic substrates indicates the presence of reducible monosaccharides of CNC and advanced levels of hydrolytic breakdown [65]. Then, the obtained sample subjected to high-speed blending revealed that the microfibers had more fractured fibers due to the intense shearing forces applied during the blending process. It showed the milky white color in Figure 5e. Thereafter, the sample was processed through ultrasonication, changing to a semi-white color with good dispersion of microfibers (Figure 5f). Finally, the obtained sample from the centrifuged process showed a translucent and good suspension of CNCs from DSM (Figure 5g). Figure 5h–k shows the end product of CNCs from TSG, DSM, BL, and BS, indicating a similar appearance.

### 3.5. Morphological Analysis

The surface morphology of four different untreated raw bamboo fiber samples (TSG, DSM, BL, and BS) was investigated by SEM analysis (Figure 6). The microphotographic images of raw samples showed a long stick-like nature at the width size of 18–300 μm (Figure 6a,c,e,g). After the cellulose conversion by the bleaching process, cellulose showed the fragment-irregular-fiber shape of 2–45 μm, which can be seen in Figure 6b,d,f,h. However, after treatment with sodium hydroxide and sodium chlorite solutions, the cellulose fibers decreased in thickness and length.

The FE-SEM-STEM micrographs show a nanorod-like structure of CNCs from TSG, DSM, BL, and BS (CNC-TSG, CNC-DSM, CNC-BL, and CNC-BS) (Figure 7a–d). The morphological structures of CNC were similar. The CNC-TSG and CNC-DSM had rod shapes with a diameter of 11–49 nm and a length of 100–232 nm. The CNC-BL and CNC-BS showed roughly rod-like morphology with a diameter of 16–62 nm and a length of 86–375 nm. Interfacial hydrogen bonding appeared to be responsible for the aggregation of bamboo particles, resulting in the formation of larger particles. The aggregation tendency of CNC-TSG and CNC-DSM was more apparent and could be attributed to their lower negative surface charge compared to CNC-BL and CNC-BS [61]. This observation is consistent with the particle size distribution of CNCs from different bamboo species, as depicted in Figure 8.

### 3.6. The Particle Size Analysis of Four Different CNCs

The statistical distribution of particles in CNC suspensions prepared from four bamboo species was determined using DLS. However, this distribution serves as an indication of the potential aggregate presence and as an assessment of the level of size polydispersity. The size obtained from DLS is based on the diffusion coefficient of particles, which is converted into a hydrodynamic radius using the Stokes–Einstein equation. Therefore, the size determined by this method represents the radius of a sphere that exhibits the same diffusion coefficient as the rod-like CNCs [5]. The particle size from Figure 8a CNC-TSG has the size of the particles, which reaches a peak at 68.1 nm, accounting for 24.6%, having an average particle size of 74.4 nm. Figure 8b shows that the particle size of CNCs from DSM is found to be 68.1 nm, accounting for 25.2%, with the average particle size at 77.2 nm. Similarly, in CNC-BL, as shown in Figure 8c, the particle size peak at 91.3 nm, accounting for 24.7%, with an average particle size of 100.8 nm. Then, Figure 8d shows the particle size CNC-BS is 68.1 nm, accounting for about 26%, and an average particle size of 78.5 nm. This standard Maxwell–Boltzmann distribution was evidence of CNC synthesis as more than 90% of the volume fraction of particle size, which was confirmed in the nm range. Based on the DLS evaluation of four different bamboo samples, the smallest average particle size was obtained from TSG, and the largest particle size was revealed in BL. The images of FE-SEM-STEM well support these results. However, differences were observed in the sizes, which is due to the interaction between the CNCs and the water molecules (Figure 7). These images were confirmed by the rod-like morphology of all synthesized CNC bamboo samples [44]. The results of different sizes of synthesized CNCs depended on the bamboo species, and these different sizes of CNCs were related to several factors. Generally, the fiber length distributions of one, three, and five-year bamboos consist of 1.6–3.1 mm. By contrast, current investigations have a fiber length as long as 6.4 mm. Compared to one-year-old bamboos, three and five-year-old bamboos exhibited a greater proportion of fibers measuring less than 1.6 mm. Moreover, the average fiber length varied among the horizontal layers of the bamboos at different ages [24]. Specifically, the outer layer had considerably shorter fibers than the middle and inner layers, indicating that bamboos have a significant concentration of short fibers in their outermost layer.

### 3.7. Tensile Strength and Elongation at Break Analysis of Nanocomposite Films

Figure 9 shows the TS and EB of the cassava starch-CNC-DSM nanocomposite films. The TS of the cassava starch-CNC-DSM nanocomposite films was dependent on the amount of CNC-DSM. The maximum TS of 4.2 MPa was observed for the composite film with the addition of 0.4 g CNC. Similarly, Nasution et al. reported that the cassava starch-based film with 0.4 CNC showed a TS value of 4.1 MPa [66]. However, our study showed that the optimal amount of 0.4 CNC resulted in a better TS value due to the improved dispersion and interactions between CNCs and the cassava starch polymeric matrix. At higher amounts of CNC in the nanocomposite films, TS decreased because of CNC aggregation (Figure 9a). Agustin et al. [18] reported that the entanglement of CNCs in a polymer matrix becomes more dense, leading to agglomeration and forming voids or defects in the material. These voids and defects act as stress concentrators, weakening the material and decreasing TS. On the other hand, the orientation and alignment of CNCs lead to less uniformity, leading to a decrease in TS. At lower amounts of 0.2 g CNC, the nanocomposite film showed a higher EB value at 55.04% than that of those films, as shown in Figure 9b. At higher amounts of CNC, the EB decreased, indicating an increase in the stiffness of the nanocomposite films. This was due to the restriction of the chain stretching [33].

### 3.8. Solubility and Swelling Analysis of Cassava Starch-CNC-DSM Nanocomposite Films

Figure 10a shows the solubility of the cassava starch-CNC-DSM nanocomposite films. When CNCs were added to cassava starch-based films, the solubility of the cassava starch film decreased as the content of CNCs increased. This decrease in solubility of the nanocomposite films was due to the formation of hydrogen bonds between the CNCs and the starch molecules, reducing the mobility of the polymer chains and limiting the starch’s ability to dissolve in a solvent. Similarly, Ma et al. [33] report that the increased content of CNC had the effect of linearly decreasing the water solubility of the films. The effect of CNCs on the swelling of cassava starch-based films is shown in Figure 10b. The swelling of the starch polymer is dependent on the amount of CNC. The addition of a low CNC content (0.2 g) into cassava starch-based film increased swelling. This swelling may be indicative of fewer interactions between the cassava starch molecules and the CNC molecules. However, the addition of higher CNC contents (0.4 and 0.6 g) slightly decreased the swelling of cassava starch-based films. This can be described that in the amount of 0.4 and 0.6 g CNC, the presence of the highly crystalline CNCs reduces the mobility of the starch polymer chains and limits their ability to absorb solvent, leading to a decrease in the swelling percentage. In addition, the strong interactions between CNC molecules and cassava starch molecules decrease space intermolecularly, which limits water absorption and penetration [67]. This effect is related to forming a network structure within the polymer, which impedes solvent penetration into the polymer matrix [33].

### 3.9. Water Vapor Permeability (WVP) Analysis of Cassava Starch-CNC-DSM Nanocomposite Films

As shown in Figure 11, the addition of CNCs into the cassava starch-based films reduced the WVP because CNCs act as a filler in the polymer matrix, forming a lengthier tortoise pathway for absorption and diffusion of water vapor. The CNCs interact with the starch molecules through hydrogen bonding, which promotes the formation of a more compact and dense structure in the polymer. This more compact structure limits the movement of water molecules through the material, resulting in a decrease in WVP [68]. More importantly, the high aspect ratio and small size of the CNC provide a tortuous path for water molecules to diffuse through the material, further reducing the WVP of the polymer. In addition, the high crystallinity of the CNC causes physical barriers that prevent the permeability of water vapor through the films [69]. In the amount of 0.6 g CNC, the film somewhat lost the WVP compared to the film with 0.4 g of CNC. The slight increment in WVP is related to the higher number, larger size, and poor dispersion of CNCs [33].

### 3.10. FTIR Analysis of Cassava Starch-CNC-DSM Nanocomposite Films

Figure 12 shows the FTIR spectral analysis of CNC-DSM, cassava starch, and cassava starch-CNC-DSM nanocomposite films (0.2, 0.4, and 0.6 g). The FTIR spectrum of CNC-DSM showed broad and shoulder peaks at 3316 cm and 2895 cm^− 1^ that are attributed to the stretching of –OH and C–H groups, respectively. A narrow peak observed at 1650 cm^−1^ was attributed to the H–O–H stretching vibration, corresponding to water adsorbed by cellulose molecules. Additionally, the peak observed at around 1425 cm^−1^ corresponded to the –CH_2_ scissoring bending vibration, while the C–O–C stretching vibration produced a band at 1200–1000 cm^−1^. Finally, the C–H rocking vibration contributed to the band observed at 1000–800 cm^−1^ [33]. The pure cassava starch film displayed several absorption peaks in its FTIR spectrum. The stretching frequency of the –OH group was responsible for a broad absorption peak observed at 3280 cm^−1^, while the C–H stretching vibration produced a peak at 2925 cm^−1^. A strong absorption peak at 1650 cm^−1^ confirmed the presence of water. Additionally, peaks at 1410 and 1340 cm^−1^ were assigned to –CH_2_ bending in the plane and C–OH bending vibration, respectively. The C–O–C antisymmetric bridge stretching contributed to the peak observed at 1149 cm^−1^, while C–O–H bending vibration produced a peak at 1010 cm^−1^ [31,70]. Different amounts of CNC-DSM added to the cassava starch-based films of FTIR spectra did not show a major difference in the functional groups. Noticeably, the peaks of 1025 and 895 cm^−1^ are associated with distinct cellulose peaks arising in all nanocomposite films. Noticeably, the peak intensity at 3280 cm^−1^ corresponding to the –OH group decreased with the increase of 0.4 and 0.6 CNC contents, which confirms the strong interactions between CNCs and cassava starch molecules [69]. The interaction causes polymer chains to lose hydroxyl groups in the form of water molecules, and, as this water is evaporated in the drying process, the amount of –OH groups tends to decrease. The starch polymer matrix was reinforced with CNCs by forming a network of entangled fibers via hydrogen bonding, resulting in an increasing TS in Figure 9. On the other hand, the peak intensity of –OH groups (3280 cm^−1^) increased with the adding 0.2 g of CNC, which may be indicative of fewer interactions between CNCs and cassava starch molecules. This results in high free –OH groups [71].

### 3.11. Atomic Force Microscopy Analysis of Nanocomposite Films

Figure 13 shows typical topographic images of cassava starch-CNC-DSM composite film surfaces containing various amounts of CNC obtained using AFM. The images revealed that the CNC particles were uniformly dispersed on the surface of the 0.2 g and 0.4 g CNC-containing nanocomposite films, indicating a better interaction between the matrix and CNCs, and confirming the enhancement of the films’ mechanical properties. Additionally, increasing the CNC content in the composite film surface resulted in increased roughness, from 0.7 to 2.5 nm, due to the high aspect ratio and large surface area of the CNCs, which increased the surface energy. However, as the CNC content increased to 0.6 g, the AFM images exhibited uneven surfaces with high roughness, indicating the CNC agglomerates’ presence. The stress experienced by the films became concentrated on these aggregates, leading to a decrease in TS in Figure 9 [72].

### 3.12. Contact Angle of Cassava Starch-CNC-DSM Nanocomposite Films

Figure 14 shows the dynamic contact angle of cassava starch-CNC-DSM nanocomposite films. The water dynamic contact angle of cassava starch-CNC-DSM nanocomposite films decreased with time from 0 to 60 s. The dynamic contact angle of nanocomposite films increased as the number of CNCs increased. The increase in the contact angle represents a rise in the hydrophobicity of the films. This implies that the cassava starch-CNC nanocomposite films are less hydrophilic than cassava starch films. The strong bond between the chains of CNC and cassava starch may be one important factor in the increase in contact angle, confirmed by FTIR spectra (Figure 12). In addition, CNCs reduce the free space between the molecules of cassava starch films, thus limiting the absorption of water [67]. Another reason for the increased water contact angle of nanocomposite films was an increase in surface roughness when the number of CNCs increased, as confirmed by the AFM image (Figure 13) [73]. According to Slavutsky and Bertuzzi, the hydrophobic character of starch films was improved by the addition of CNCs from sugarcane bagasse [68]. This contact angle result is consistent with the solubility (Figure 10) of cassava starch-CNC-DSM nanocomposite films.

## 4. Conclusions

This study utilized the chemical–mechanical method to prepare CNCs from four bamboo species. The results indicated that the extracted CNC samples from DSM exhibited a slightly higher yield and crystallinity index than TSG, BL, and BS. However, the crystallite size of CNC-DSM was quite bigger than that of TSG, BL, and BS. All samples showed nano-sized rod-like structures. The addition of CNC-DSM improved the water barrier properties of the cassava starch-based films. Moreover, the incorporation of 0.4 g CNC-DSM in cassava starch nanocomposite film resulted in excellent tensile strength. This was related to the smaller diameter and higher aspect ratio, improving the mechanical properties of polymeric nanocomposites. Based on these findings, it is suggested that cassava starch-CNC-DSM nanocomposite films can be utilized in food packaging applications.

## Figures and Tables

**Figure 1 polymers-15-02622-f001:**
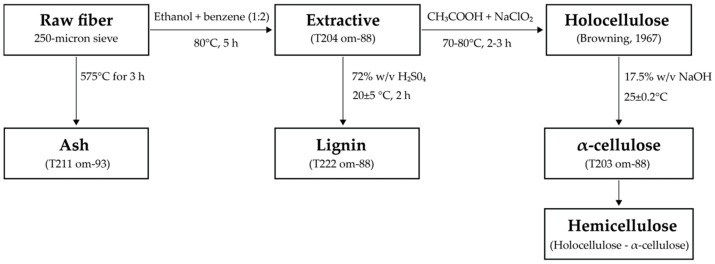
The chemical constituent analysis diagram of raw fiber [36].

**Figure 2 polymers-15-02622-f002:**
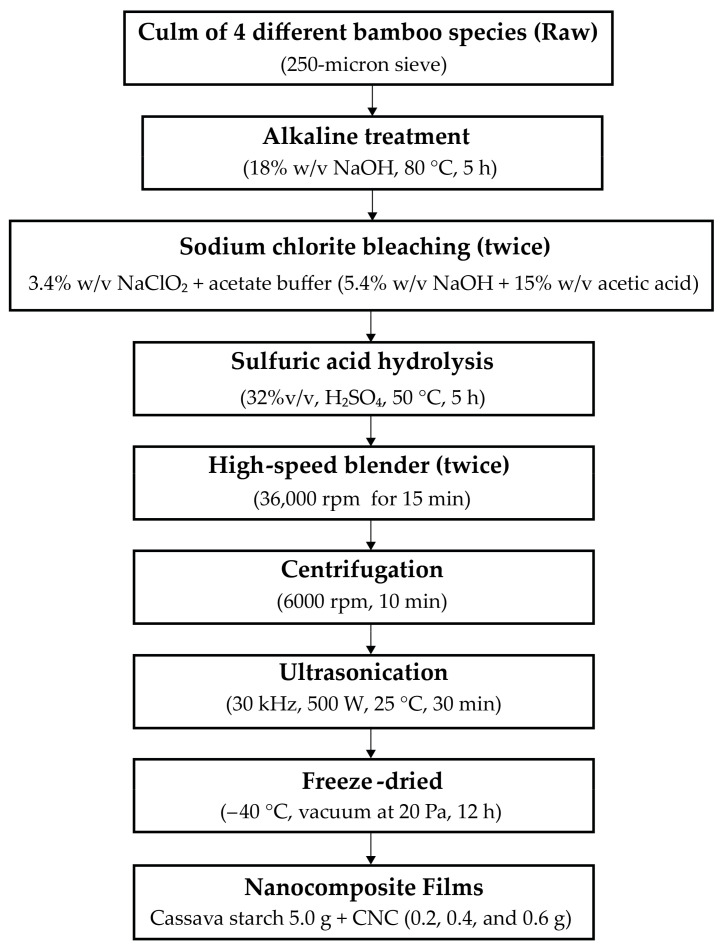
Experimental design of CNC extraction from different bamboo fibers and preparation of nanocomposite films.

**Figure 3 polymers-15-02622-f003:**
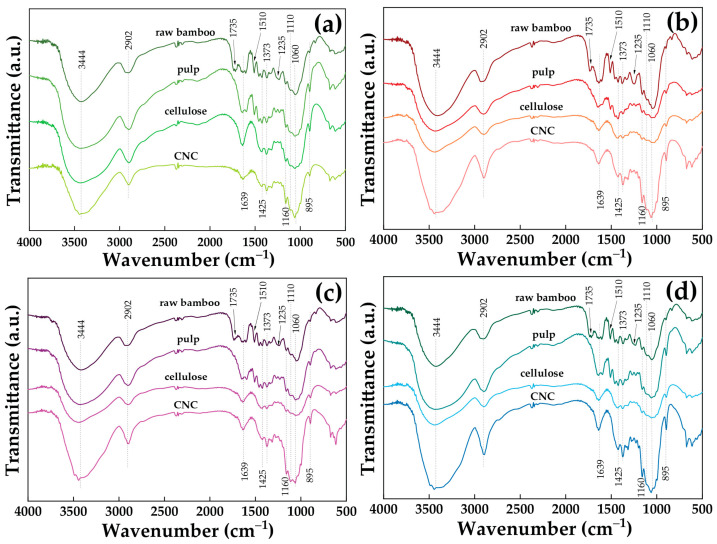
FTIR spectral analysis of raw bamboos, pulps, cellulose, and CNC: (**a**) TSG, (**b**) DSM, (**c**) BL, and (**d**) BS.

**Figure 4 polymers-15-02622-f004:**
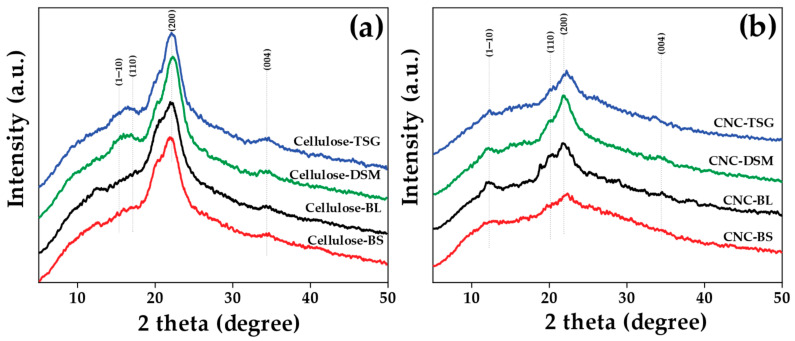
XRD analysis: (**a**) cellulose and (**b**) CNC of four different species of bamboo.

**Figure 5 polymers-15-02622-f005:**
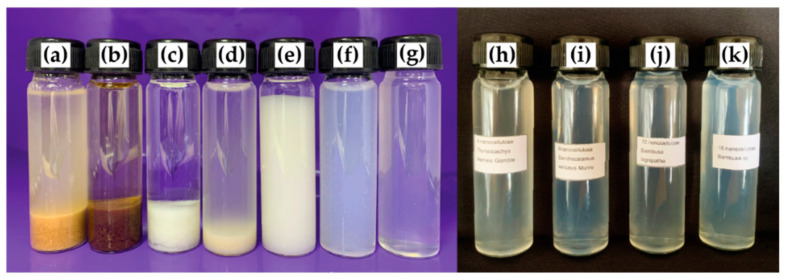
Samples in different stages of DSM: (**a**) raw bamboos, (**b**) alkaline treatment, (**c**) bleached, (**d**) acid hydrolyzed, (**e**) blended at high-speed, (**f**) ultrasonicated, (**g**) centrifuged, (**h**) CNC-TSG, (**i**) CNC-DSM, (**j**) CNC-BL, and (**k**) CNC-BS.

**Figure 6 polymers-15-02622-f006:**
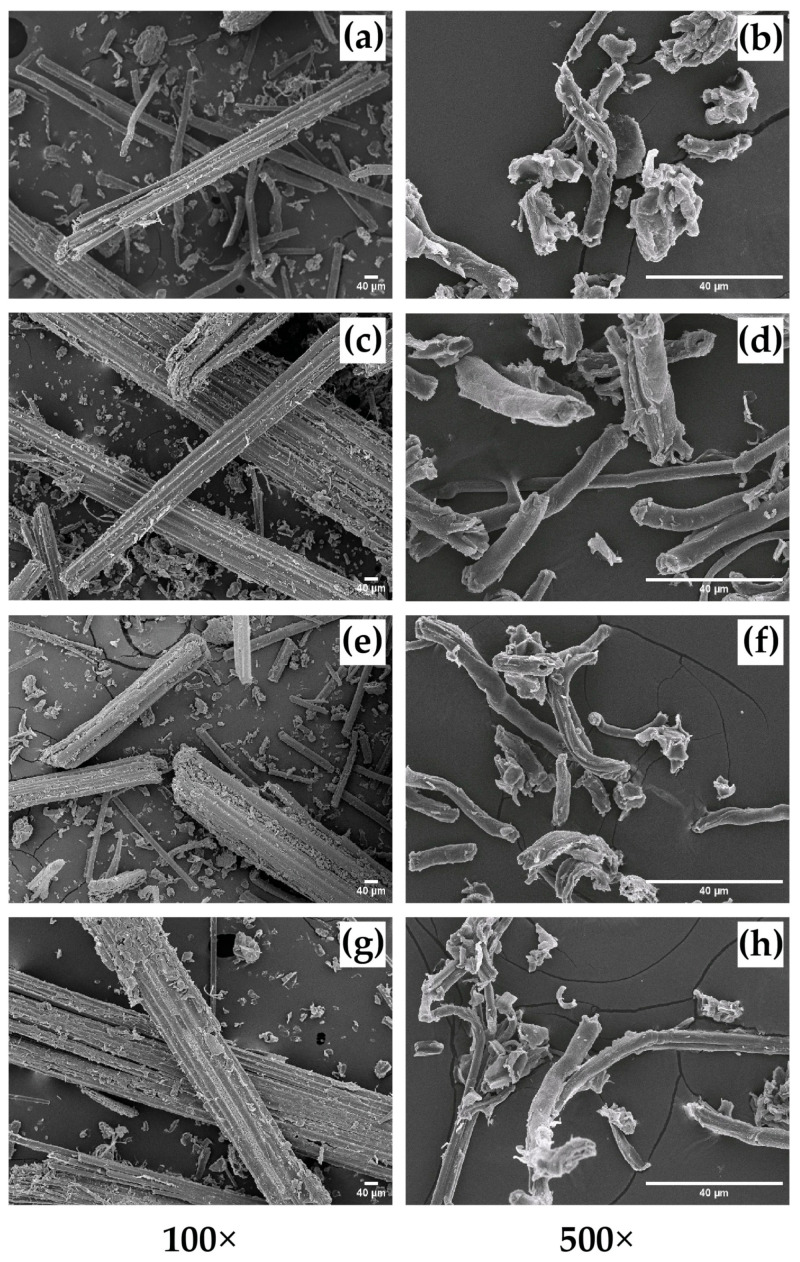
SEM analysis of raw bamboos and cellulose of four different bamboo species: (**a**) raw bamboo TSG (**b**) cellulose TSG (**c**) raw bamboo DSM, (**d**) cellulose. DSM (**e**) raw bamboo BL (**f**) cellulose BL (**g**) raw bamboo BS, and (**h**) cellulose BS.

**Figure 7 polymers-15-02622-f007:**
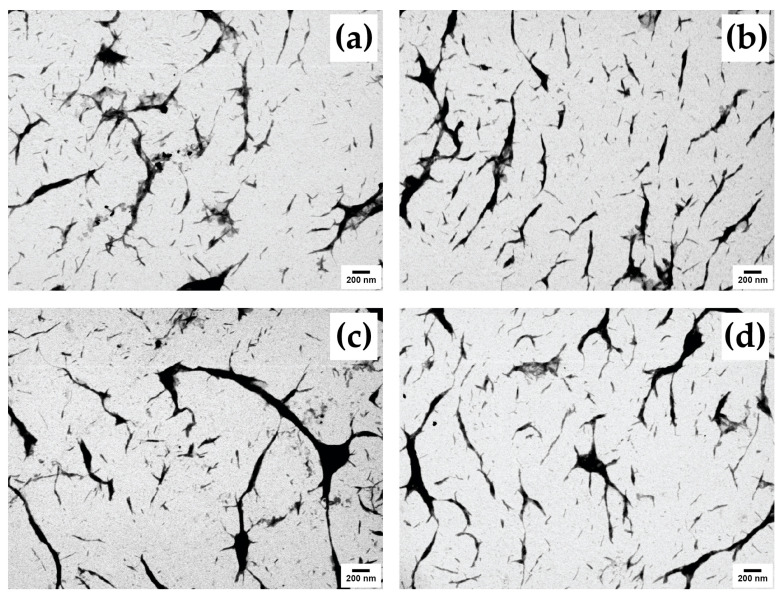
FE-SEM-STEM micrographs of CNCs from four different bamboo species 30,000× magnification: (**a**) CNC-TGS, (**b**) CNC-DSM, (**c**) CNC-BL, and (**d**) CNC-BS.

**Figure 8 polymers-15-02622-f008:**
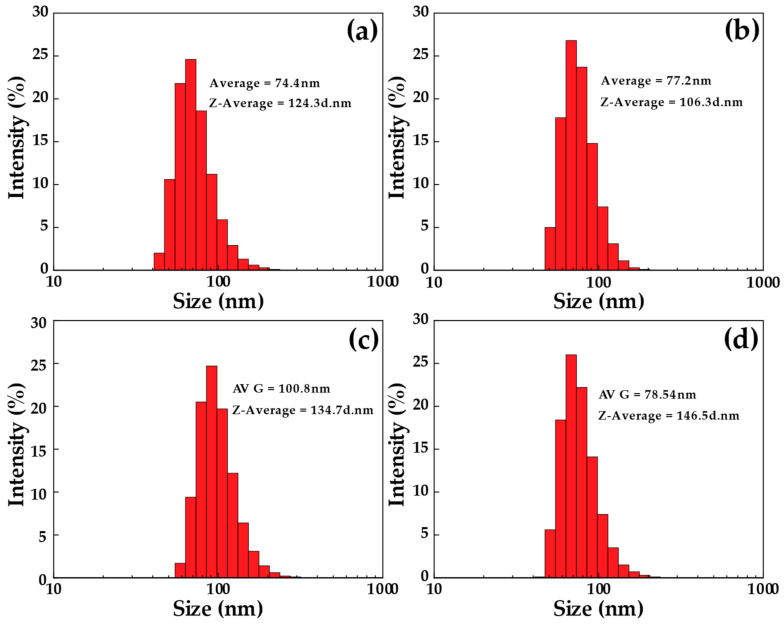
DLS analysis: (**a**) CNC-TSG, (**b**) CNC-DSM, (**c**) CNC-BL, and (**d**) CNC-BS.

**Figure 9 polymers-15-02622-f009:**
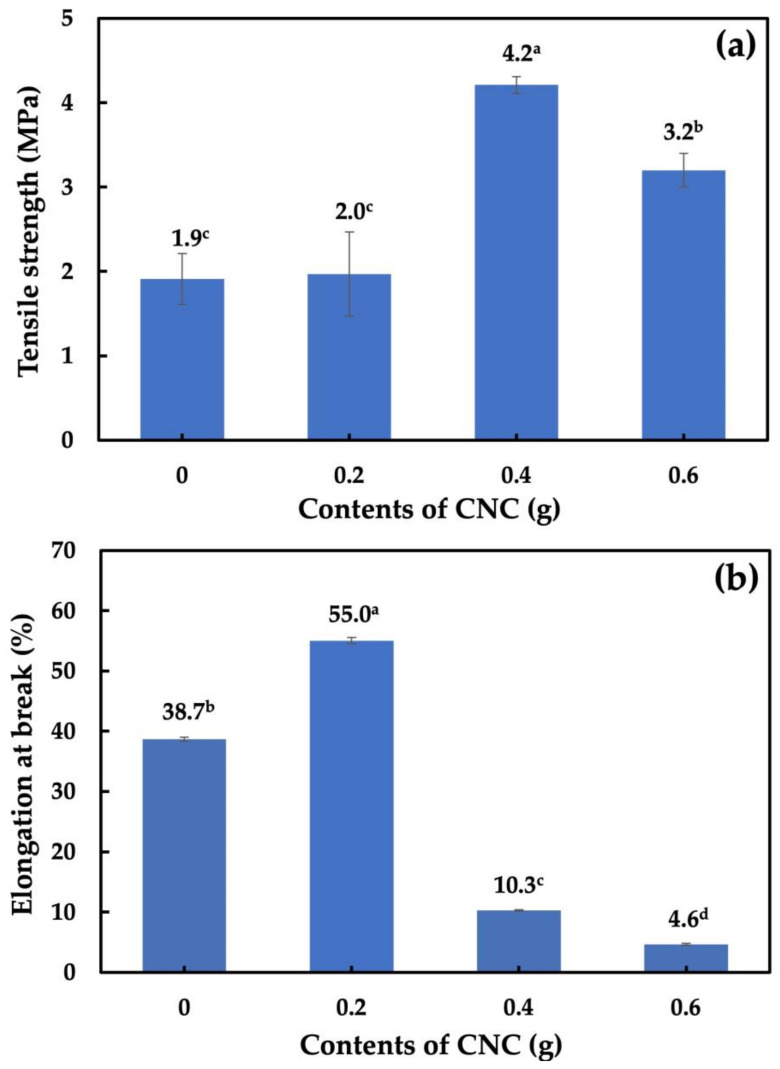
(**a**) Tensile strength and (**b**) Elongation at break of cassava starch-CNC-DSM nanocomposite films. Note: values indicated with the same letters are not significantly different at *p* ≤ 0.05.

**Figure 10 polymers-15-02622-f010:**
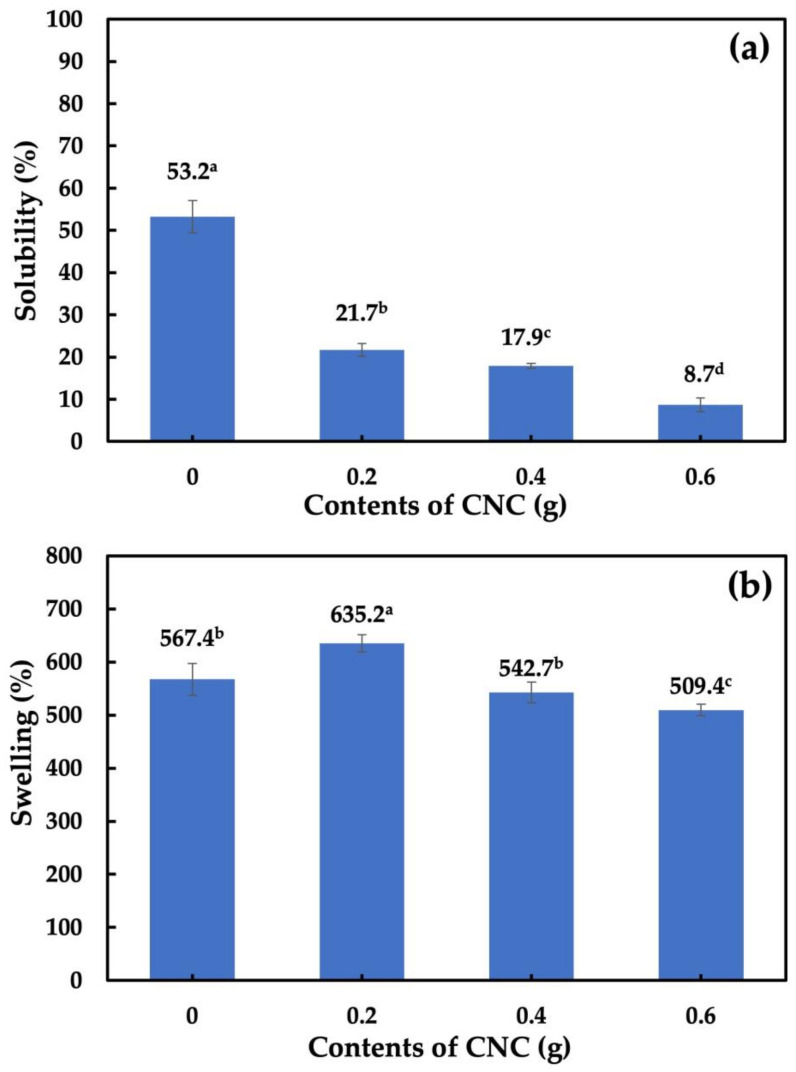
Solubility (**a**) and swelling (**b**) of nanocomposite films: Cassava starch, 0.2 g CNC-DSM, 0.4 g CNC-DSM, and 0.6 g CNC-DSM. Note: values indicated with the same letters are not significantly different at *p* ≤ 0.05.

**Figure 11 polymers-15-02622-f011:**
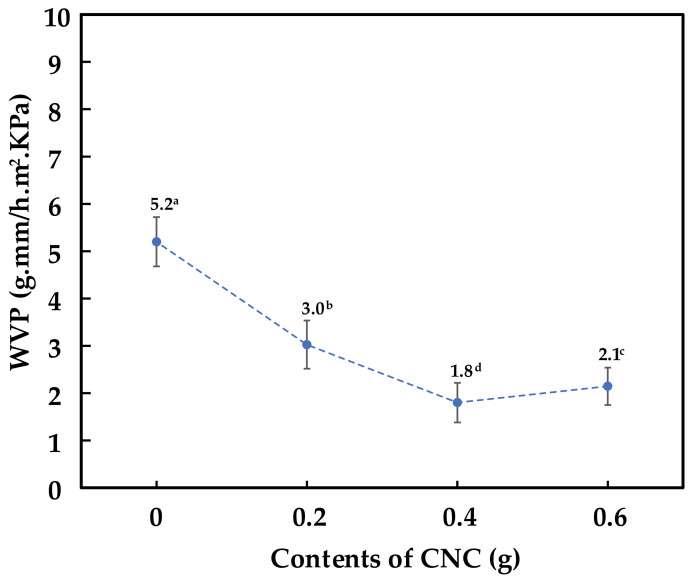
Water vapor permeability (WVP) analysis: Cassava starch, 0.2 g CNC-DSM, 0.4 g CNC-DSM, and 0.6 g CNC-DSM composite film. Note: values indicated with the same letters are not significantly different at *p* ≤ 0.05.

**Figure 12 polymers-15-02622-f012:**
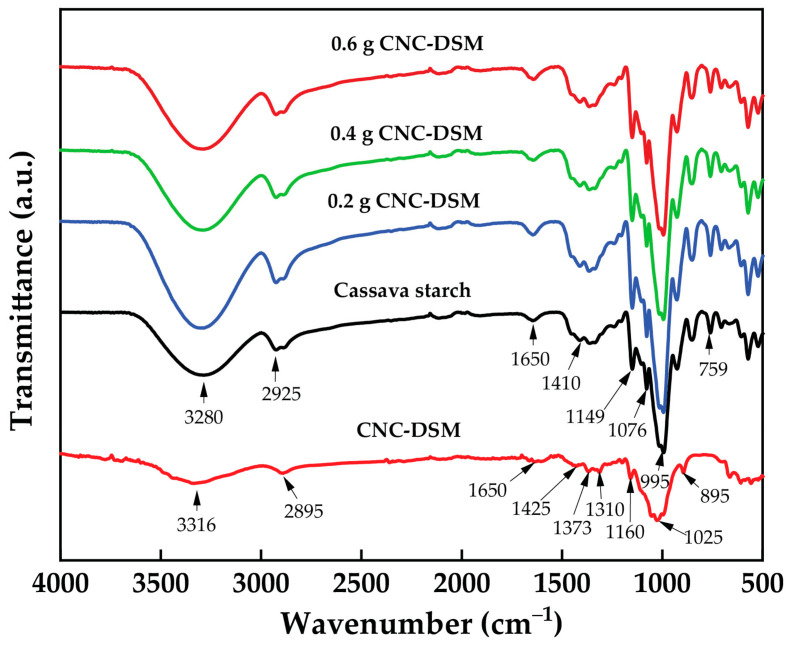
FTIR analysis of CNC, cassava starch, and cassava starch-CNC-DSM nanocomposite films.

**Figure 13 polymers-15-02622-f013:**
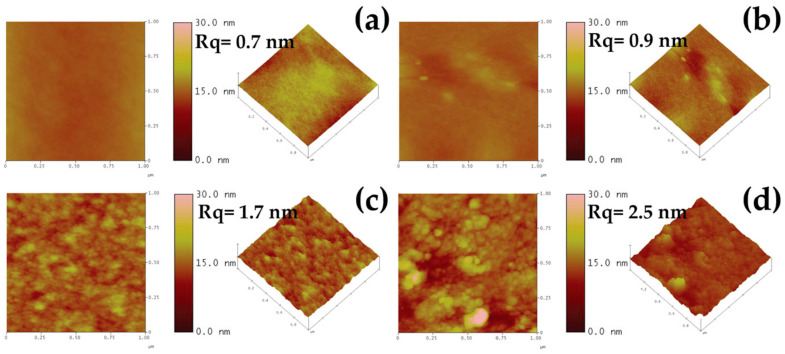
AFM image in 2D and 3D phase of nanocomposite films: (**a**) Cassava starch, (**b**) 0.2 g CNC-DSM, (**c**) 0.4 g CNC-DSM, and (**d**) 0.6 g CNC-DSM.

**Figure 14 polymers-15-02622-f014:**
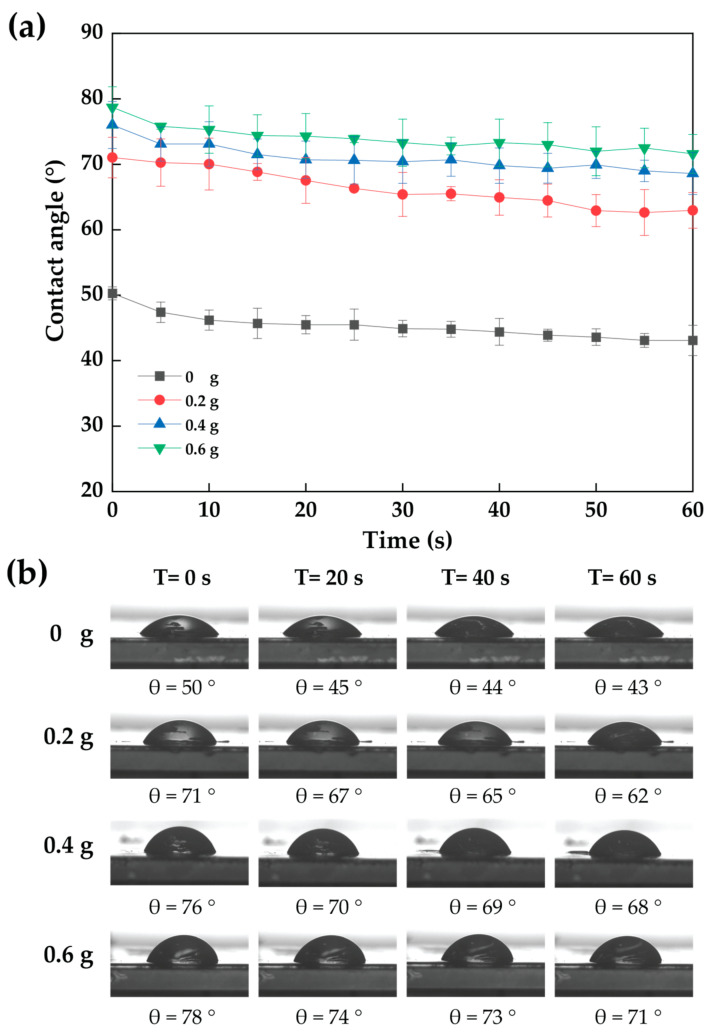
Dynamic contact angle measurement of cassava starch-CNC-DSM nanocomposite films (**a**) and water contact angle images droplet on the surface of the cassava starch-CNC-DSM nanocomposite films with times (**b**).

**Table 1 polymers-15-02622-t001:** Chemical constituents of four bamboo species.

Samples	Holocellulose (%)	α-Cellulose (%)	Hemicellulose (%)	Lignin (%)	Extractive (%)	Ash (%)
TSG	70.6 ± 0.6	40.2 ± 0.3	30.4 ± 0.6	22.5 ± 0.6	3.1 ± 0.1	2.4 ± 0.4
DSM	74.6 ± 0.6	42.5 ± 0.9	32.1 ± 0.8	23.4 ± 06	4.3 ± 0.4	1.9 ± 0.1
BL	73.1 ± 0.8	41.7 ± 1.1	31.4 ± 1.5	28.9 ± 0.6	3.2 ± 0.1	2.0 ± 0.1
BS	64.4 ± 0.9	37.4 ± 0.8	27.0 ± 1.1	24.4 ± 0.7	4.9 ± 0.1	2.3 ± 0.2

**Table 2 polymers-15-02622-t002:** The crystallinity index and yield of cellulose and CNC of four different bamboo species.

Species of Bamboo	Crystallinity Index (%)		Crystallite Size Perpendicular to Plane 200 (nm)		Yield (%)	
	Cellulose	CNC	Cellulose	CNC	Cellulose	CNC
TSG	45.9	39.7	2.9	1.3	40.8	32.9
DSM	46.3	42.2	2.8	1.6	42.1	33.9
BL	45.4	40.7	2.1	1.5	41.5	31.4
BS	45.4	38.9	2.3	1.4	40.1	30.5

## Data Availability

The data presented in this study are available on request from the corresponding author. The data are not publicly available due to privacy.

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
