# Peer review of "Preparation and Characterization of Cellulose Nanocrystals from Bamboos and Their Application in Cassava Starch-Based Film"

_polymers, 2023, doi:10.3390/polym15122622_

Round 1
Reviewer 1 Report
The manuscript entitled "Preparation and Characterization of Cellulose Nanocrystals from Bamboos and Its Application in Cassava Starch-Based Film" reported a chemical-mechanical method of combining different species of bamboo (Thyrsostachys siamesi Gamble, Dendrocalamus sericeus Munro (DSM), Conversion of cellulose from Bambusa logispatha, and Bambusa sp.) to cellulose nanocrystals (CNC). In general, it is an interesting and valuable topic to deserving a research article.
However, there are still many problems to be solved. So this reviewer would suggest a major revision before its acceptance.
1. Overall the draft is good but needs more careful editing.
2. Some proper nouns and terms should be further explained in the Introduction section of this paper, and some research progress should be elaborated in more detail.
3. The relevant research background of bamboo is less introduced. It should be mentioned in the article what is the advantage of this bamboo, not only in Southeast Asia, but also whether the bamboo in other regions has relevant properties.
4. All figures should have the text highlighted in them bolded to make them clearer, and the colors used in the images should be higher contrast colors.
5. More introduction on the structure, properties and applications of CNC should be provided with some recent supporting articles: A mixed acid methodology to produce thermally stable cellulose nanocrystal at high yield using phosphoric acid; Packaging and degradability properties of polyvinyl alcohol/gelatin nanocomposite films filled water hyacinth cellulose nanocrystals; etc.
6. Equations must be typeset using appropriate fonts and layout. Excel figures must not contain the superfluous outer frame. The same information must not be duplicated in Tables and Figures.
7. It is suggested that the author make a good flow chart to illustrate the relevant ideas of the experiment.
8. Some formulas in this paper are suggested to be rewritten by software.
9. The literatures cited in this manuscript are too short of timeliness. The reviewer suggests quoting more recent literatures and comparing the data in this manuscript: Industrial Crops and Products 173 (2021) 114103; Chinese Chemical Letters (2021) https://doi.org/10.1016/j.cclet.2021.03.
10. There are still some typos and grammar issues. In addition, please carefully check the references to ensure the full information is included.
Author Response
Reviewer 1
Comments and Suggestions for Authors
The manuscript entitled "Preparation and Characterization of Cellulose Nanocrystals from Bamboos and Its Application in Cassava Starch-Based Film" reported a chemical-mechanical method of combining different species of bamboo (Thyrsostachys siamesi Gamble, Dendrocalamus sericeus Munro (DSM), Conversion of cellulose from Bambusa logispatha, and Bambusa sp.) to cellulose nanocrystals (CNC). In general, it is an interesting and valuable topic to deserving a research article.
However, there are still many problems to be solved. So this reviewer would suggest a major revision before its acceptance.
- Overall the draft is good but needs more careful editing.
Answer:
Thank you for your comment. We have carefully edited our manuscript.
- Some proper nouns and terms should be further explained in the Introduction section of
this paper, and some research progress should be elaborated in more detail.
Answer:
As you suggest, explanations of proper nouns and terms have been provided in the Introduction section. At the same time, relevant research progress has been described and included in the revised manuscript. It now reads on page 2 lines 47-94.
- The relevant research background of bamboo is less introduced. It should be mentioned in the article what is the advantage of this bamboo, not only in Southeast Asia but also whether the bamboo in other regions has relevant properties.
Answer:
Thank you for your comment. We have included the detailed relevant research background of bamboo, its advantages, and its properties in the revised version of the manuscript. It now reads on page 2 lines 47-59.
- All figures should have the text highlighted in them bolded to make them clearer, and the colors used in the images should be higher contrast colors.
Answer:
Thank you for your valuable comment. We have refined all figure text and contrast colors in the revised version of the manuscript.
- More introduction on the structure, properties and applications of CNC should be provided with some recent supporting articles: A mixed acid methodology to produce thermally stable cellulose nanocrystal at high yield using phosphoric acid; Packaging and degradability properties of polyvinyl alcohol/gelatin nanocomposite films filled water hyacinth cellulose nanocrystals; etc.
Answer:
Thank you for your comment. We have explained the structure, properties, and applications of CNC and included suggested references in appropriate places in the revised version of the manuscript. It now reads on page 2 ref. no. 25,26, etc.(23,24).
- Equations must be typeset using appropriate fonts and layout. Excel figures must not contain the superfluous outer frame. The same information must not be duplicated in Tables and Figures.
Answer:
As you suggested, all equations have been properly formatted, and the Excel figure's outer frame layout has been removed. The data duplicated in tables and figures have been revised in the revised version of the manuscript.
- It is suggested that the author make a good flow chart to illustrate the relevant ideas of the experiment.
Answer:
As you suggested, we have included the flow chart of the experimental details in the revised manuscript. It now reads on page 5 lines 185-186 and Figure 1.
- Some formulas in this paper are suggested to be rewritten by software.
Answer:
As you suggested, we have revised the formulas using software in the revised manuscript. It now reads on pages 4 and 6 lines 176, 216, 220, 239, and 250.
- The literatures cited in this manuscript are too short of timeliness. The reviewer suggests quoting more recent literatures and comparing the data in this manuscript: Industrial Crops and Products 173 (2021) 114103; Chinese Chemical Letters (2021) https://doi.org/10.1016/j.cclet.2021.03.
Answer:
As you suggested, we have included the reference in the revised manuscript.
It now reads on page 2 ref. no. 9.
https://doi.org/10.1016/j.cclet.2021.03. This reference DOI number is incorrect; we could not add this reference in the revised manuscript.
- There are still some typos and grammar issues. In addition, please carefully check the references to ensure the full information is included.
Answer:
As you suggested, we have carefully checked the typos, grammar mistakes, and reference information. These corrections have been included in the revised version of the manuscript.

Reviewer 2 Report
The presented manuscript is dedicated to the production of cellulose from various types of bamboo flour and the further production of a composite material with starch.
In my opinion, the works that contain detailed characteristics of cellulose and the raw materials from which it is obtained are more reliable and allow a complete picture of the study.
The authors paid attention to this in their work, which makes it more attractive.
In the theoretical part, the authors pay little attention to previously published results, for example, in the work https://aip.scitation.org/doi/10.1063/5.0015669, characteristics of CNC reinforced cassava starch biocomposites is considered.
I recommend that the authors update the theoretical part, taking into account a deeper analysis of the published results.
Another example is https://doi.org/10.1016/j.carbpol.2014.03.049.
The purpose of the work is not explicit and it is better to reformulate it.
If the first part of the experimental part leaves a positive impression, then the second part of the manuscript raises a lot of questions.
So I don't understand why mechanics data for all compositions is not given?
Here I recommend the authors to round the values to tenths, otherwise it is required to provide data on the accuracy of measurements and the methods used.
The conclusions in the manuscript are very concise and require revision, taking into account the questions and recommendations below.
Line 24. "pre-treated" - maybe the authors have to enter "pre-treated"?!
Lines 107-109. This statement confuses me a little. In my understanding, a-cellulose is a fraction of cellulose that is insoluble in an alkaline solution, i.e. holocellulose minus heteropolymers...
Line 92. Unfortunate expression, I propose to change it to the authors.
2.3. Why drying was carried out at different temperatures?
3.3. XRD and percent yield of cellulose and CNC of four different bamboo species. It is necessary to check the plane indices for polymorph II. I also recommend that authors estimate the size of crystallites and provide data in the manuscript.
Figure 6. The quality of the drawings needs to be improved.
Lines. 434-435. I do not agree that the area is increasing ...
Lines 437-439. I don't quite understand how CNC crystals swell in water?!
3.8. Solubility and swelling analysis of cassava starch-CNC-DSM composite films. This section needs to be completely redone!
The presented manuscript is dedicated to the production of cellulose from various types of bamboo flour and the further production of a composite material with starch.
In my opinion, the works that contain detailed characteristics of cellulose and the raw materials from which it is obtained are more reliable and allow a complete picture of the study.
The authors paid attention to this in their work, which makes it more attractive.
In the theoretical part, the authors pay little attention to previously published results, for example, in the work https://aip.scitation.org/doi/10.1063/5.0015669, characteristics of CNC reinforced cassava starch biocomposites is considered.
I recommend that the authors update the theoretical part, taking into account a deeper analysis of the published results.
Another example is https://doi.org/10.1016/j.carbpol.2014.03.049.
The purpose of the work is not explicit and it is better to reformulate it.
If the first part of the experimental part leaves a positive impression, then the second part of the manuscript raises a lot of questions.
So I don't understand why mechanics data for all compositions is not given?
Here I recommend the authors to round the values to tenths, otherwise it is required to provide data on the accuracy of measurements and the methods used.
The conclusions in the manuscript are very concise and require revision, taking into account the questions and recommendations below.
Line 24. "pre-treated" - maybe the authors have to enter "pre-treated"?!
Lines 107-109. This statement confuses me a little. In my understanding, a-cellulose is a fraction of cellulose that is insoluble in an alkaline solution, i.e. holocellulose minus heteropolymers...
Line 92. Unfortunate expression, I propose to change it to the authors.
2.3. Why drying was carried out at different temperatures?
3.3. XRD and percent yield of cellulose and CNC of four different bamboo species. It is necessary to check the plane indices for polymorph II. I also recommend that authors estimate the size of crystallites and provide data in the manuscript.
Figure 6. The quality of the drawings needs to be improved.
Lines. 434-435. I do not agree that the area is increasing ...
Lines 437-439. I don't quite understand how CNC crystals swell in water?!
3.8. Solubility and swelling analysis of cassava starch-CNC-DSM composite films. This section needs to be completely redone!
Author Response
Reviewer 2
Comments and Suggestions for Authors
The presented manuscript is dedicated to the production of cellulose from various types of bamboo flour and the further production of a composite material with starch.
In my opinion, the works that contain detailed characteristics of cellulose and the raw materials from which it is obtained are more reliable and allow a complete picture of the study.
The authors paid attention to this in their work, which makes it more attractive.
In the theoretical part, the authors pay little attention to previously published results, for example, in the work https://aip.scitation.org/doi/10.1063/5.0015669, characteristics of CNC reinforced cassava starch biocomposites is considered.
Answer:
Thank you for your comment. We have included suggested references in the appropriate places, and the previously published results have been discussed in the theoretical part of the revised version of the manuscript. It now reads on page 13 ref. no. 59 line 453-456.
I recommend that the authors update the theoretical part, taking into account a deeper analysis of the published results.
Another example is https://doi.org/10.1016/j.carbpol.2014.03.049.
Answer:
As you suggested, we have included the reference in the revised manuscript. It now reads on page 15 ref. no. 62 lines 496-499.
The purpose of the work is not explicit and it is better to reformulate it.
If the first part of the experimental part leaves a positive impression, then the second part of the manuscript raises a lot of questions.
So I don't understand why mechanics data for all compositions is not given?
Answer:
Thank you for your valuable comment. The morphology of all samples is similar and has a better yield of DMS. Hence, we chose only one of the DMS for filmmaking.
Here I recommend the authors to round the values to tenths, otherwise it is required to provide data on the accuracy of measurements and the methods used.
Answer:
Thank you for your valuable suggestion, we have revised the values to tenths in the revised manuscript.
The conclusions in the manuscript are very concise and require revision, taking into account the questions and recommendations below.
Line 24. "pre-treated" - maybe the authors have to enter "pre-treated"?!
Answer:
Thank you for your comment. The required change has been carried out in the revised manuscript. It now reads on page 1 line 27.
Lines 107-109. This statement confuses me a little. In my understanding, a-cellulose is a fraction of cellulose that is insoluble in an alkaline solution, i.e. holocellulose minus heteropolymers...
Answer:
Thank you for your comment. The required change has been carried out in the revised manuscript. It now reads on page 3 lines 135-138.
Line 92. Unfortunate expression, I propose to change it to the authors.
Answer:
Thank you for your comment. The required change has been carried out in the revised manuscript. It now reads on page 3 lines 115 and 121-122.
2.3. Why drying was carried out at different temperatures?
Answer:
The drying process was conducted at the same temperature of 85 oC. The typo error has been corrected in the revised manuscript. It now reads on page 3 line 135.
3.3. XRD and percent yield of cellulose and CNC of four different bamboo species. It is necessary to check the plane indices for polymorph II. I also recommend that authors estimate the size of crystallites and provide data in the manuscript.
Answer:
As you suggested. We have calculated the CNC crystallite size, discussed it, and included it in the revised manuscript. It now reads on pages 9-10 lines 346-349 and table 2.
Figure 6. The quality of the drawings needs to be improved.
Answer:
As you suggested, we have increased the quality of Figure 6 in the revised manuscript.
Lines. 434-435. I do not agree that the area is increasing ...
Answer:
Thank you for your valuable suggestion. We have corrected and refined this section in the revised manuscript. It now reads on page 14 lines 470-488.
Lines 437-439. I don't quite understand how CNC crystals swell in water?!
Answer:
Thank you for your valuable suggestion. We have corrected the statement and refined it in the revised manuscript. It now reads on page 14 lines 470-488.
3.8. Solubility and swelling analysis of cassava starch-CNC-DSM composite films. This section needs to be completely redone!
Answer:
Thank you for your valuable suggestion. We have repeated the experiment with cassava starch-CNC-DSM composite films solubility and swelling analysis. The obtained data have been included in the revised manuscript [Figure: 9]. In addition, we conducted the contact angle test [Figure: 13], and the data are discussed in the revised manuscript. It now reads on pages 14-15 lines 470-488.

Round 2
Reviewer 1 Report
Accept in present form
Author Response
Reviewer 1
Comments and Suggestions for Authors
Accept in present form
Answer:
Thank you very much for accepting our manuscript for publication in Polymers.

Reviewer 2 Report
The authors made a number of edits to the manuscript, but it still contains a huge number of errors, typos, inaccuracies, etc.
I recommend the authors to carry out a thorough work with the manuscript and only after that submit the work for review.
The following are examples of errors, etc. (not a complete list!):
Line 69. "ca be" needs to be corrected.
Line 83. "the CNC is biodegradability" - needs to be checked.
Lines 103, 103. In my opinion it is better to indicate the type of bamboo, so that the reader has an idea of ​​the previous studies.
Line 214. "follows [86]:" - The specified link is not in the list!?
Lines 225, 226. "Tensile strength (TS) and elongation at break (EB) of nanocomposite films reinforced with CNC." -?!
Line 228. "small pieces" - better to use - strips. "nanocomposite film" - can be removed.
Line 234. Replace "0.0500" with "0.05"
Line 254. "as described by Thanakkasaranee et al." delete.
Lines 260-264. Re-information! You need to check and leave in one place.
Figure 3. The quality of the diffractograms is not very good. When processing them, it is possible to obtain incorrect results regarding the size of crystallites, etc.
Line 370. "Figure 3c" - there is no such pattern!
Line 69. "ca be" needs to be corrected.
Line 83. "the CNC is biodegradability" - needs to be checked.
Lines 225, 226. "Tensile strength (TS) and elongation at break (EB) of nanocomposite films reinforced with CNC." -?!
Line 228. "small pieces" - better to use - strips. "nanocomposite film" - can be removed.
Author Response
Reviewer 2
Comments and Suggestions for Authors
The authors made a number of edits to the manuscript, but it still contains a huge number of errors, typos, inaccuracies, etc.
I recommend the authors to carry out a thorough work with the manuscript and only after that submit the work for review.
The following are examples of errors, etc. (not a complete list!)
Thank you so much for your comment.
Line 69. "ca be" needs to be corrected.
Answer: This sentence was corrected. It now reads on page 2 lines 69-70.
“CNC can be extracted from various natural fibers such as sugarcane bagasse [15], walnut shell [16], palm oil empty fruit bunch [17], garlic stalks [18], and bamboo [19-21].”
Line 83. "the CNC is biodegradability" - needs to be checked.
Answer: This sentence was corrected. It now reads on page 2 lines 82-83.
“In addition, CNC is a biodegradable, lightweight, low-density, and transparent mate-rial [27,28].”
Lines 103, 103. In my opinion it is better to indicate the type of bamboo. so that the reader has an idea of ​​the previous studies.
Answer: The type of bamboo was added. It now reads on page 3 lines 102-104.
“Until today, several studies have reported the preparation of CNC from bamboo fibers, but in most cases, it is only one species of bamboo such as Phyllostachys hetero-cycle, Dendrocalamus, or Pseudosasa amabilis [20,21,35].”
Line 214. "follows [86]:" - The specified link is not in the list!?
Answer: Reference No. 40 was added. It now reads on page 6 lines 217-219.
“The ICr is the crystalline phase's integrated intensity, whereas Inon-Cr is the non-crystalline phase's integrated intensity as a background as follows [40]:”
40 Nam, S.; French, A.D.; Condon, B.D.; Concha, M. Segal crystallinity index revisited by the simulation of X-ray diffraction patterns of cotton cellulose Ibeta and cellulose II. Carbohydr Polym 2016, 135, 1-9, doi:10.1016/j.carbpol.2015.08.035.
Lines 225, 226. "Tensile strength (TS) and elongation at break (EB) of nanocomposite films reinforced with CNC." -?!
Answer: This sentence was corrected. It now reads on page 6 line 230-232.
“The tensile tester (Model MCT-1150, Japan) was used to determine the tensile strength (TS) and elongation at break (EB) of the cassava starch-CNC-DSM nanocomposite films at a crosshead speed of 10 mm/min.”
Line 228. "small pieces" - better to use - strips. "nanocomposite film" - can be removed.
Answer: This sentence was corrected. It now reads on page 6 line 232-234.
“The films were cut into 5 mm × 50 mm. All samples were equilibrated in the desiccator at 50±10% RH 25 °C for 48 h and were analyzed according to the JISK-6251-7.”
Line 234. Replace "0.0500" with "0.05"
Answer: We have replaced "0.05" in the revised manuscript. It now reads on page 6 line 237-238.
“The dried sample was weighed at 0.05 g initial dry weight.”
Line 254. "as described by Thanakkasaranee et al." delete.
Answer: We have deleted "as described by Thanakkasaranee et al." in the revised manuscript. It now reads on page 6 line 257-258.
“The water vapor permeability (WVP) of nanocomposite films was measured according to the ASTM standard method (ASTM, E96-93, 1993).”
Lines 260-264. Re-information! You need to check and leave it in one place.
Answer: We have deleted the information content and provided only section “2.6. Characterizations” in the revised manuscript. It now reads on page 5 lines 194-198.
“In addition, the FTIR spectrometer in ATR mode was used to examine the structural connections of nanocomposite films reinforced with CNC. The film samples were cut into small pieces (10 mm × 10 mm), then placed in the sample container. The spectra were run using a scan rate of 64, in the wavenumber of 4,000−500 cm-1.”
Figure 3. The quality of the diffractograms is not very good. When processing them, it is possible to obtain incorrect results regarding the size of crystallites, etc.
Answer: We have provided the better quality diffractograms in Figure 3. It now reads on page 9 lines 360-361.
Line 370. "Figure 3c" - there is no such pattern!
Answer: Figure 3c was changed to Figure 4c. It now reads on page 10 lines 367-368.
“Figure 4c showed a white color appearance due to the bleaching process.”

Round 3
Reviewer 2 Report
The authors have made corrections according to the previous review, but at the moment the manuscript is not ready for publication and contains many inaccuracies and errors. Authors should not be guided only by the provided reviews, but it is also possible to independently check their work with the involvement of third-party specialists or editors).
Below is a list of current comments and suggestions:
When using the term "a-cellulose", the Greek letter alpha must be used.
Lines 118-120. Check out this sentence!
Lines 138, 139. The sentence is badly formulated incorrectly)!!!
Line 173. China. delete point.
Fig.1. "sie ve" is an extra space.
Lines 214, 215. The sentence needs to be corrected!
Line 225. Equation 3 is wrong!
Table 1. Need to check and fix the table!
3.2. FTIR Analysis. Instead of the term "peak" it is better to use "band".
Lines 118-120. Check out this sentence!
Lines 138, 139. The sentence is badly formulated incorrectly)!!!
Line 173. China. delete point.
Fig.1. "sie ve" is an extra space.
Lines 214, 215. The sentence needs to be corrected!
Author Response
Answer to Reviewer 2
comments and Suggestions for Authors
The authors have made corrections according to the previous review, but at the moment the manuscript is not ready for publication and contains many inaccuracies and errors. Authors should not be guided only by the provided reviews, but it is also possible to independently check their work with the involvement of third-party specialists or editors).
Below is a list of current comments and suggestions:
When using the term "a-cellulose", the Greek letter alpha must be used.
Thank you so much for your comment.
Answer: We have used " α-cellulose" instead of the term "a-cellulose". It now reads on pages 3, 7, and 9 lines 130, 135, 138, 307, 311, 312, 314, and 388.
Lines 118-120. Check out this sentence!
Answer: We have corrected the sentence as you commented. It now reads on page 3 lines 118-119.
Lines 138, 139. The sentence is badly formulated incorrectly)!!!
Answer: We have revised the sentence as you commented. It now reads on page 3 lines 135-140.
Line 173. China. delete point.
Answer: We have corrected it as you commented. It now reads on page 4 line 190.
Fig.1. "sie ve" is an extra space.
Answer: We have corrected it as you commented. It now reads on page 5 line 204.
Lines 214, 215. The sentence needs to be corrected!
Answer: The sentence has been revised as you commented. It now reads on page 6 lines 233-234.
Line 225. Equation 3 is wrong!
Answer: Equation 3 have revised as you commented. It now reads on page 6 line 244.
Table 1. Need to check and fix the table!
Answer: As you commented, we have checked and confirmed that the values in the Table 1 were corrected. It now reads on page 7.
3.2. FTIR Analysis. Instead of the term "peak" it is better to use "band".
Answer: We have used “band” instead of the term “peak” as you commented. It now reads on pages 7, 8, 9,12,16, and 17 lines 326, 338, 339, 340, 342, 344, 348, 353, 362, 372, 373, 380, 465, 469, 573, and 582.
Round 4
Reviewer 2 Report
Table 1. If the authors sum up the values for alpha cellulose, hemicellulose, lignin, extractive and ash, do they get values greater than 100%?!
Line 335 onwards. The term "peak" is used to describe diffractograms. The authors have vainly replaced it with "band".
Fig.2. How do the authors explain the increase in the intensity of the 895 cm-1 band for CNC when this band is known to correspond to the amorphous phase. For the TSG sample, the situation is reversed. The obtained spectra must be correlated with the XRD data. How do the authors explain the low values for CNC compared to cellulose? What values of the crystallinity index are obtained from the IR spectra?
Table 2. Crystallite sizes should be checked and compared with already published data.
Conclusions. The conclusions presented need to be expanded. It is also not entirely clear, if the crystallinity of cellulose is higher than that of CNC, then why not use it as an additive? Such an additive can provide better barrier properties, etc.
Line 335 onwards. The term "peak" is used to describe diffractograms. The authors have vainly replaced it with "band".
Author Response
Reviewer
Comments and Suggestions for Authors
Dear Reviewer
We would like to thank Reviewer 2 for taking the necessary time and effort to review our manuscript for 4 times. We sincerely appreciate all your valuable comments and suggestions, which helped us in improving the quality of our manuscript.
Table 1. If the authors sum up the values for alpha cellulose, hemicellulose, lignin, extractive, and ash, do they get values greater than 100%?!
Answer: The analysis begins with the extraction of extractive using ethanol-benzene solvent according to T204 om-88. Then, the holocellulose was analyzed from extractive-removed fiber using the acid chlorite method according to Browning. The α-cellulose was extracted from the holocellulose according to T203 om-88 and the hemicellulose was calculated by the difference of holocellulose and α-cellulose. The lignin was analyzed from extractive-removed fiber according to T222 om-88. Ash was analyzed according to T211 om-93. In general, the extractive, lignin, and ash contents are not included in the calculation of holocellulose, α-cellulose, and hemicellulose contents. In addition, ash content is analyzed using a different method from that of the analysis of raw material from the initial fiber. However, the sum of chemical constituents obtained from holocellulose, extractive, and lignin contents was close to 100%, in which this result ranges from 94-105% (the content was over 100% due to triplication and SD of analysis). Our result was similar to other research that has been recorded in previous studies by Rusch et al. [12], Mu et al. [45], and Pacaphol et al. [37].
As you commented, therefore, we have checked and added ± stand error in Table 1. We strongly confirm that the data in Table 1 was corrected. We have shown the chemical constituent analysis diagram of raw fiber in Figure 1. It now reads on pages 3-4 lines 140-147, which provides convenience and readability for readers.
- Rusch, F.; Wastowski, A.D.; de Lira, T.S.; Moreira, K.C.C.S.R.; de Moraes Lucio, D. Description of the component properties of species of bamboo: a review. Biomass Conversion and Biorefinery 2023, 13, 2487-2495.
- Mu, B.; Tang, W.; Liu, T.; Hao, X.; Wang, Q.; Ou, R. Comparative study of high-density polyethylene-based biocomposites reinforced with various agricultural residue fibers. Industrial Crops and Products 2021, 172, 114053.
- Pacaphol, K.; Aht-Ong, D. Preparation of hemp nanofibers from agricultural waste by mechanical defibrillation in water. Journal of Cleaner Production 2017, 142, 1283-1295, doi:10.1016/j.jclepro.2016.09.008.
Chemical constituents (%) calculation:
|
Chemical constituents (%) |
Samples |
1 |
2 |
3 |
AVG |
SD |
|
Holocellulose (%) |
TSG |
70.26 |
70.26 |
71.26 |
70.59 |
0.58 |
|
DSM |
73.95 |
74.95 |
75.02 |
74.64 |
0.60 |
|
|
BL |
72.36 |
73.96 |
73.00 |
73.11 |
0.81 |
|
|
BS |
63.48 |
64.58 |
65.25 |
64.44 |
0.89 |
|
|
α-cellulose (%) |
TSG |
40.51 |
39.97 |
40.22 |
40.23 |
0.27 |
|
DSM |
42.05 |
43.53 |
42.01 |
42.53 |
0.86 |
|
|
BL |
41.04 |
41.05 |
43.01 |
41.70 |
1.13 |
|
|
BS |
36.85 |
38.42 |
37.05 |
37.44 |
0.85 |
|
|
Hemicellulose (%) |
TSG |
29.75 |
30.29 |
31.04 |
30.36 |
0.65 |
|
DSM |
31.90 |
31.42 |
33.01 |
32.11 |
0.81 |
|
|
BL |
31.32 |
32.91 |
29.99 |
31.40 |
1.46 |
|
|
BS |
26.63 |
26.17 |
28.19 |
27.00 |
1.06 |
|
|
Lignin (%) |
TSG |
23.21 |
22.34 |
22.02 |
22.52 |
0.61 |
|
DSM |
22.92 |
23.22 |
24.02 |
23.39 |
0.57 |
|
|
BL |
29.35 |
28.22 |
29.00 |
28.86 |
0.58 |
|
|
BS |
24.37 |
25.15 |
23.79 |
24.44 |
0.68 |
|
|
Extractive (%) |
TSG |
2.99 |
3.20 |
3.06 |
3.08 |
0.11 |
|
DSM |
4.04 |
4.15 |
4.78 |
4.32 |
0.40 |
|
|
BL |
3.24 |
3.12 |
3.09 |
3.15 |
0.08 |
|
|
BS |
5.06 |
4.93 |
4.82 |
4.94 |
0.12 |
|
|
Ash (%) |
TSG |
3.60 |
3.02 |
3.73 |
3.45 |
0.38 |
|
DSM |
1.89 |
1.77 |
2.00 |
1.89 |
0.12 |
|
|
BL |
2.06 |
2.02 |
1.92 |
2.00 |
0.07 |
|
|
BS |
2.11 |
2.48 |
2.30 |
2.30 |
0.19 |
Line 335 onwards. The term "peak" is used to describe diffractograms. The authors have vainly replaced it with "band".
Answer: The term "peak" is used to describe diffractograms as you commented. It now reads on page 8 (line 327-347),17 (line 773-789), and 18 line 812
Fig.2. How do the authors explain the increase in the intensity of the 895 cm-1 band for CNC when this band is known to correspond to the amorphous phase. For the TSG sample, the situation is reversed. The obtained spectra must be correlated with the XRD data. How do the authors explain the low values for CNC compared to cellulose? What values of the crystallinity index are obtained from the IR spectra?
Answer: The peak at 895 cm-1 is β-glucosidic linkages between sugar units that are the C1–H de-formation of cellulose. As the size of CNC decreases, the relative proportion of amorphous regions might change, potentially affecting the intensity of the FTIR vibrational band at 895 cm-1.
We cannot calculate the crystallinity index from IR used for the corresponding functional group analysis.
Table 2. Crystallite sizes should be checked and compared with already published data.
Conclusions. The conclusions presented need to be expanded.
Answer: Thank you so much for your valuable comment. We have checked the crystallite sizes of cellulose and CNC from previously published works. It now reads on pages 9-10, lines 381-441 reference No. 23, 54, 55, 56, and 57. In addition, we have checked the XRD graph and re-calculated the crystallite sizes of cellulose and CNC.
The calculation of crystallite size.
Scherrer Equation: D = Kl/bcosq
|
Samples |
Peak position(2theta) |
FWHM |
Beta |
Cos Theta |
Crystallite sizes |
|
Cellulose-TSG |
22.02131 |
2.95144 |
0.05151 |
0.98159 |
2.86402 |
|
Cellulose-DSM |
22.14460 |
2.97511 |
0.05193 |
0.98139 |
2.84183 |
|
Cellulose-BL |
21.67948 |
4.00192 |
0.06985 |
0.98216 |
2.11101 |
|
Cellulose-BS |
21.66298 |
3.60796 |
0.06297 |
0.98218 |
2.34145 |
|
CNC-TSG |
21.88286 |
6.70653 |
0.11705 |
0.98182 |
1.26011 |
|
CNC-DSM |
21.76228 |
5.15121 |
0.08991 |
0.98202 |
1.64025 |
|
CNC-BL |
21.26611 |
5.60663 |
0.09785 |
0.98283 |
1.50578 |
|
CNC-BS |
21.76115 |
6.23832 |
0.10888 |
0.98202 |
1.35441 |
For example, calculation: D = Kl/bcosq
k = 0.94, λ = 0.15406 nm, and β is the full width at half maximum of 200 reflections. The unit degree was changed to radian.
D = 0.94 x 0.15406
0.0515 x 0.98159
D = 2.86402 nm
Answer: We have expanded the conclusions as you commented. It now reads on page 20 lines 867-868
It is also not entirely clear, if the crystallinity of cellulose is higher than that of CNC, then why not use it as an additive? Such an additive can provide better barrier properties, etc
Answer: Not only crystallinity, the particle size or crystallite size strongly affects the properties of polymer materials or films. The CNC typically exhibits nanoscale dimensions with a larger surface area compared to the larger size of that cellulose. The smaller size of crystallite such as that of CNC with a larger surface area generally enhances the mechanical, barrier, and chemical properties of the native polymer like cassava starch films due to strong physical interaction at the interphase between CNC and cassava starch regarding smaller size and larger surface area than cellulose with the larger size.
